# The Effects of Chemically Modified Biochar on Biomethane Production from Glucose and Sugar Beet Pulp

**DOI:** 10.3390/ma18071608

**Published:** 2025-04-02

**Authors:** Julia K. Nowak, Joanna Rosik, Kacper Szadziński, Marvin T. Valentin, Katarzyna E. Kosiorowska, Andrzej Białowiec, Sylwia Stegenta-Dąbrowska, Kacper Świechowski

**Affiliations:** 1Faculty of Biology and Animal Science, Wrocław University of Environmental and Life Sciences, 38c Chełmońskiego St., 50-375 Wrocław, Poland; 117535@student.upwr.edu.pl; 2Institute of Environmental Engineering, Wrocław University of Environmental and Life Sciences, Grunwaldzki Sq. 24, 50-363 Wrocław, Poland; 3Department of Applied Bioeconomy, Wrocław University of Environmental and Life Sciences, 51-630 Wrocław, Poland; 120323@student.upwr.edu.pl (K.S.); or m.valentin@bsu.edu.ph (M.T.V.); katarzyna.kosiorowska@upwr.edu.pl (K.E.K.); andrzej.bialowiec@upwr.edu.pl (A.B.); sylwia.stegenta-dabrowska@upwr.edu.pl (S.S.-D.); kacper.swiechowski@upwr.edu.pl (K.Ś.); 4Department of Agricultural and Biosystems Engineering, Benguet State University, Km. 5, La Trinidad, Benguet 2601, Philippines

**Keywords:** biochar, chemically activated biochar, anaerobic digestion, sugar beet pulp, glucose

## Abstract

The research aimed to study the effects of straw-derived biochar and two types of chemically modified biochar on biomethane production from glucose as a model substrate and sugar beet pulp as a real substrate. The biochar chemical modification with H_3_PO_4_ acid and KOH base resulted in a change in biochar surface area properties and its functional group’s abundance and a decrease in biochar mass yield production. The anaerobic digestion process was performed in batch reactors kept at 37 °C for 20 days. The substrate-to-inoculum ratio by volatile solids was 0.5, while the mass of added biochar corresponded to 16 g·L^−1^. The results showed that neither the addition of biochar nor the chemically modified biochar had any positive effects on biomethane production or its kinetics in the case of both substrates. The highest methane production was found in reactors without biochar added, respectively, 385 and 324 mL·g_VS_^−1^ for glucose and sugar beet pulp. It is hypothesized that the anaerobic digestion process was performed under optimal conditions, and therefore, biochar could not enhance methane production. Additionally, biochar may have adsorbed some volatile fatty acids, making them less available to anaerobic microorganisms.

## 1. Introduction

The anaerobic digestion (AD) process enables the utilization of organic waste by converting it into biogas, a renewable energy source primarily composed of methane (CH_4_) and carbon dioxide (CO_2_) [1]. Anaerobic digestion technology demonstrates a positive net energy production rate, and the biomethane gas generated through this process has the potential to serve as a viable substitute for fossil fuels. When managed appropriately, AD systems pose no adverse effects on human health or the environment [2]. Despite the numerous benefits of AD, poor operational stability remains a significant barrier to its effective use [3]. Enhancing the rate of anaerobic digestion or increasing biogas yield is, therefore, a key focus in both scientific research and industrial applications. Various strategies and techniques have been explored to improve AD efficiency: co-digestion [4], pretreatment techniques [5], use of nanoparticles [6], or optimization of operational parameters for specific substrates [7]. 

Some studies have demonstrated that the addition of biochar can positively influence AD by improving microbial activity, buffering capacity, and electron transfer. For example, Ma et al. (2018) [8] reported that the addition of biochar derived from rice husks enhanced methane production by 25% due to its high surface area and porosity, which facilitated microbial colonization. Unfortunately, most of the previous literature focused on the role of pristine biochar in AD, when modified biochar has also shown promise in optimizing anaerobic digestion. To improve the application of biochar in AD systems, many studies have improved the physicochemical properties of biochar by various functional methods [9]. For instance, functionalization with metal oxides such as Fe_3_O_4_ or MnO_2_ has been found to accelerate direct interspecies electron transfer (DIET), an essential mechanism for efficient methane production [10]. Wang et al. (2023) [11] observed that magnetite-loaded biochar enhanced the methane production rate by 1.6-fold compared to the control. Moreover, alkaline-activated biochar has been shown to improve buffering capacity, reducing the risk of process inhibition caused by pH fluctuations [12]. However, numerous studies revealed that biochar addition may not enhance biomethane production and may even have an adverse effect on AD performance [13]. The contradictory results indicate that the influence of biochar on AD science is not yet developed, and more studies are required to explain the factors affecting the AD process.

In this study, potassium hydroxide (KOH) and phosphoric acid (H_3_PO_4_) were selected as chemical modifiers due to their proven effectiveness in enhancing the physicochemical properties of biochar relevant to anaerobic digestion. KOH activation is known to increase the specific surface area and porosity of biochar and introduce alkaline functional groups, which can improve the material’s buffering capacity and promote the adsorption of inhibitory compounds such as ammonia and volatile fatty acids (VFAs) [14]. In contrast, H_3_PO_4_ modification introduces phosphate groups and creates acidic functional sites, which can enhance cation exchange capacity, improving structural stability and development of mesopores [15]. The contrasting properties imparted by these two modifiers present an opportunity to evaluate how surface chemistry and functionalization influence the performance of biochar in anaerobic digestion processes.

The significance of this research lies in its comprehensive evaluation of chemically modified biochars, which have been less frequently studied in the context of anaerobic digestion processes compared to unmodified biochars. This study analyzes both straw-derived and chemically modified biochars with distinct surface functionalization to elucidate their impact on biomethane production under controlled conditions. Furthermore, by utilizing both a model substrate (glucose) and a real substrate (sugar beet pulp), this research seeks to enhance understanding of the role of biochar in various anaerobic digestion scenarios. This methodology contributes to closing the knowledge gap regarding the mechanisms by which biochar properties influence methane yield. The critical properties of modified biochar that influence anaerobic digestion performance are not yet fully understood, which restricts its broader application in AD systems [9]. Consequently, it is essential to regulate biochar properties and develop modified biochar. The objective of this study is to systematically evaluate the effects of straw-derived biochar and two types of modified biochar on biomethane production derived from glucose and sugar beet pulp. Through the analysis of key parameters, including biomethane yield, biochar surface properties, the content of functional groups on the biochar surface, and the accumulation of volatile fatty acids (VFAs), this research aims to elucidate the mechanisms through which biochar enhances methanogenic fermentation.

## 2. Materials and Methods

This study utilized six materials: one inoculum, two substrates (glucose and sugar beet pulp), one biochar, and two chemically modified biochars as additives in the anaerobic digestion process. The overall experimental workflow is illustrated in Figure 1, which outlines the sequential steps in material preparation, biochar production and modification, anaerobic digestion testing, and data analysis. 

The process began with the drying and grinding of input materials: sugar beet pulp, glucose, and wheat straw. Wheat straw was subjected to pyrolysis at 600 °C for 4 h to produce biochar. A portion of this biochar was chemically impregnated using either 30% H_3_PO_4_ (acid) or 30% KOH (base), then dried, further pyrolyzed at 500 °C for 1 h, and washed with water to obtain acid- and base-modified biochars (BC_A and BC_B, respectively). Subsequently, inoculum, glucose, and beet pulp were combined in various configurations, with or without biochar additives, for the anaerobic digestion experiments conducted using the AMPTS system. After AD, data analysis was carried out, including kinetic parameter determination and the physicochemical characterization of materials.

### 2.1. Materials

The inoculum (I) was made from the digestate collected from a 1 MW_el_ commercial agricultural biogas plant (Bio-Wat Sp. z o.o., Świdnica, Poland). The digestate was mechanically sifted through a fabric filtration bag to clean it of various solids: untreated substrate, plastics, stones, etc. The purified liquid digest (inoculum) was stored in a plastic container at room temperature for 24 h before being used as an inoculum to start the AD process.

The substrates used were glucose and sugar beet pulp. Glucose (G), with the chemical formula C_6_H_12_O_6_, is a pure chemical. It was stored in a sealed plastic container at room temperature. Sugar beet pulp (BP) in pellet form was purchased at a local market. The pellets were ground using a knife mill (LMN100, Testchem, Pszów, Poland) until a homogeneous consistency was achieved. The ground BP was then stored in a plastic bag at room temperature.

The biochar (BC) was made from wheat straw collected from a farm in Wrocław (Lower Silesia, Poland). The straw was ground to a size of <1 mm using the laboratory knife mill. The shredded wheat straw was stored in a plastic bag at room temperature until its conversion to biochar. The conversion process was as follows: the wheat straw was placed into four metal containers (sizes of container 30 cm × 9 cm × 7 cm), which were then sealed with lids. The sealed containers with materials were placed into a muffle furnace (Snol 8.1/1100, Utena, Lithuania). After two containers were placed in the furnace chamber, the chamber was closed and filled with CO_2_ gas to create an inert atmosphere. The flow rate of the inert gas was 10 mL^.^min^−1^. The furnace programmer was then set and activated. The process temperature was 600 °C, and the pyrolysis duration was 4 h. After that, the heating was turned off, and the chamber was left to cool to room temperature.

Biochar was chemically modified using two methods: with a 30% H_3_PO_4_ and with a 30% KOH solution. The chemically modified biochar was labeled BC_A (biochar modified with acid, H_3_PO_4_) and BC_B (biochar modified with base, KOH). The chemical modification procedure was as follows: 200 g of BC was placed into a 2000 mL beaker, then mixed with 300 g of both acid and base 30% solution and stirred with a glass dipstick until the solution was evenly distributed in the beaker. For 1 g of BC, 1.5 g of solution was used. After mixing, the beaker was set aside under a fume hood for 1 h. After this time, the wet chemically impregnated biochar was placed on trays and dried in a drying oven at 105 °C for 24 h until completely dry. After 24 h, the dry, chemically impregnated BC was removed from the dryer and weighed. The dried biochar was then placed in a muffle furnace and heated at 500 °C for 1 h under an inert atmosphere (CO_2_). The inert gas flow rate was 10 mL^.^min^−1^. Chemically modified biochars were then washed with water to remove the chemicals used for modification. The biochars were placed on a 63 µm mesh sieve and rinsed with warm tap water (~50 °C) for 25 min, with a 5 min break every 5 min to check the pH. The pH was measured using a pH meter (Elmetron, CPC-411, Zabrze, Poland). The rinsing process continued until the pH remained stable. The flow rate of tap water was approximately 0.216 m^3^·h^−1^, and approximately 0.82 L of water was used to wash 1 g of biochar.

### 2.2. Methods

#### 2.2.1. Substrate Analysis

The materials were analyzed to determine total solids (TSs), moisture content (MC), volatile solids (VSs), and ash content (AC). Moisture content and total solids were determined using a laboratory dryer (WAMED, model KBC-65W, Warsaw, Poland), according to PN-EN 14346:2011. Materials were placed into crucibles in triplicate for each type of material and weighed. The samples were then placed in the laboratory dryer for 24 h at 105 °C. Once the drying process was complete, the samples were weighed again. The *TS* and *MC* were calculated using the following equations: (1)$$TS=\frac{m_{wet}}{m_{dry}}\times 100,\%$$(2)MC=100−TS,%
where  TS—total solids, %; $m_{wet}$—mass of the material before the drying process, g; $m_{dry}$—mass of material after drying process at 105 °C, g; *MC*—moisture content, %.

After drying, the samples were placed in a muffle furnace (Snol 8.1/1100, Utena, Lithuania) for the combustion process. The process temperature was 550 °C, and the duration was 3 h. Once the process was complete, the crucibles were removed from the furnace, and the samples were allowed to cool to ambient temperature in a desiccator. The samples were then weighed to determine the ash content (Equation (3)) and volatile solids (Equation (4)).(3)$$AC=\frac{m_{ash}}{m_{dry}}\times 100,\%$$(4)VS=100−AC,%
where  AC—ash content, %; $m_{ash}$—mass of material after the combustion process at 550 °C, g; $m_{dry}$—mass of material after the drying process at 105 °C g; VS—organic matter content in dry matter, %.

#### 2.2.2. Determination of Carbonaceous Additive Surface Properties

Functional groups present on the surface of wheat straw, BC, BC_A, and BC_B were determined using a Nicolet iN10 integrated infrared microscope with a Nicolet iZ10 external FTIR module (Thermo Fischer Scientific, Waltham, MA, USA). For each spectrum, 32 scans were averaged over an infrared range of 400–4000 cm^−1^. The results for each material were merged to highlight the difference in functional groups due to pyrolysis and chemical modification. Next, functional group peaks were quantified as an integrated peak area. To quantify functional groups from FTIR data, the baseline for each functional group was determined as the minimum absorbance value within its wavenumber range to ensure that peak intensities were measured relative to the lowest background absorbance in the region. The following wavenumber ranges were considered: OH 3200–3600 cm^−1^; C=O 1650–1750 cm^−1^; C=C 1550–1650 cm^−1^; C-O 1000–1300 cm^−1^; and C-H bending 850–950 cm^−1^ [16]. The integrated peak area was calculated using Equation (5).(5)$$A=\int_{v_{1}}^{v_{2}}absorbance(v)dv$$
where A—integrated peak area, $v_{1}$—lower wavenumber limits for a functional group, $v_{2}$—upper wavenumber limits for a functional group, absorbance(v)—absorbance value at wavenumber.

The identification and visualization of functional group regions, including FTIR peak baseline determination and integration, were supported by AI-based computational software (ChatGPT, OpenAI, February 2024 version). The methodology involved AI-assisted numerical integration using Simpson’s Rule and graphical representation using Matplotlib in Python Version 3.9.0 (2024).

Next, the specific surface area and pore size were determined using an adsorption analyzer (ASAP 2020, Micromeritics, Atlanta, GA, USA). The analysis was performed using the volumetric method. Before measurement, the materials were degassed at 300 °C for 6 h. Nitrogen (N_2_) sorption was then measured at 77 K. Based on the obtained isotherms, the following parameters were determined: *SSA_BET_*—specific surface area calculated by BET method, *V_T_*—total pore volume < 50 nm, *L*—average pore dimension < 50 nm, *SSA_DFT_*—specific surface area calculated by Q_SDFT_ method, *V_DFT_*—pore volume by the Q_SDFT_ method, *L_DFT_*—dominant pore dimension by Q_SDFT_ method, *V*_0_—a volume of micropores by Dubinin–Radushkevich method, *L*_0_—average dimension of micropores by the Stoecklie method [17].

#### 2.2.3. The Process of Anaerobic Digestion

The experiment was conducted using two automated systems for measuring biomethane production potential (AMPTS^®^ II, BPC Instruments, Lund, Sweden). Three replications were carried out for each variant, and all variants and replications were started at the same time. Materials in the reactors were designated as follows: I—inoculum, G—glucose, BP—beet pulp, BC—biochar, BC_A—biochar modified with H_3_PO_4_ solution, BC_B—biochar modified with KOH solution. Each reactor was filled with an inoculum to a working volume of 400 mL, the total volume of one reactor being 500 mL, including 100 mL of headspace. The masses of the substrates for each reactor were measured using a laboratory spatula and then placed on plastic trays. The substrate masses were selected to ensure that the ratio of substrate volatile solids to inoculum volatile solids (SIR) was 0.5 [18]. The BC and chemically modified biochars were measured similarly, and 6.4 g of additive material was added to each reactor. The mass of added additives corresponded to 16 g_BC_·L^−1^. The weighted substrates and additives were then placed into reactors filled with inoculum. The next step before starting the AD process was to mix the contents of the reactors using a glass dipstick. The reactors were then sealed tightly and placed in a water bath. The process was carried out under mesophilic conditions (37 °C) for 20 days. During the process, the materials in the digester were mixed every hour for 2 min. The distribution of all variants is described in Table 1.

#### 2.2.4. Analysis of Volatile Fatty Acid Content and pH

During the process, samples were taken for analysis of VFA and pH. Samples were collected from each reactor on days 1, 3, 7, 14, and 20 of the process. Each variant was performed in triplicate, resulting in three samples per variant at each time point. Average values were then calculated. Samples of approximately 0.5 mL from the reactors were collected using a one-way pipette into 10 mL centrifuge tubes and then stored in a fridge at 4 °C. During sampling, the pipette tips were changed between variants to avoid cross-contamination of materials. 

Before VFA analysis, the samples were centrifuged at 10 °C at 4500 rpm for 10 min, then diluted in Milli-Q water, and re-centrifuged at 10 °C for 5 min at 15,000 rpm. Quantitative and qualitative analyses were performed using ultra-performance liquid chromatography (U-HPLC) with an UltiMate 3000 System (ThermoFisher Scientific, Cambridge, UK). A VFA standard solution (Volatile Free Acid Mix, CRM46975, Merck, Poland) with a final concentration of 10 mM was stored in a fridge at 4 °C in accordance with the manufacturer’s guidelines. The standard curve was prepared based on the standard solutions using appropriate dilutions of the analytes in Milli-Q water. The concentration of VFA (and other components such as ethanol) was determined using a HyperREZ XPCarbohydrateH + 8 μm column (Thermo Scientific, Waltham, MA, USA). Identification of analytes was performed with the UV/VIS-DAD detector at 208 ± 1 nm. To determine the possible presence of ethanol in the samples, a refractive index (RI) detector (Shodex, Ogimachi, Japan) was used simultaneously. The eluent was 0.25 mM trifluoroacetic acid, with a flow rate of 1.1 mL·min^−1^ (isocratic elution) and the column temperature set to 35 °C. Data analysis was performed using Chromeleon 7.1 Software. All samples were analyzed in triplicates. The pH was measured using a pH meter (Elmetron, CPC-411, Zabrze, Poland).

#### 2.2.5. Determination of Kinetic Parameters

After the AD process was completed, the kinetic parameters of biomethane production were determined. The data were fitted to a first-order reaction model (Equation (6)) using Statistica software (TIBCO, version 13.0, Palo Alto, CA, USA) to determine the kinetic parameters of biomethane production. Next, the biomethane production rate was determined using Equation (7).(6)$$B_{t}=B_{max}\times \left(1-e^{-k\times t}\right),mL\cdot g_{vs}^{-1}$$(7)$$r=k\times B_{max},mL\cdot {\left(g_{vs}\cdot d\right)}^{-1}$$
where $B_{t}$—biomethane obtained after time *t*, mL·g_VS_^−1^; $B_{max}$—maximum biomethane possible to obtain, mL·g_VS_^−1^; *k*—biomethane production rate constant, d^−1^; *r*—biomethane production rate, mL·(g_VS_·d)^−1^.

#### 2.2.6. Statistical Analysis of Results

The kinetic parameters (*B_max_*), (*k*), and (*r*) were subjected to ANOVA analysis to test whether there were statistically significant differences between the mean values of the results from the different variants, with a significance level of *p* < 0.05. To identify which variants showed statistically significant differences, a Tukey post-hoc test was performed. The analyses were conducted using Statistica software (TIBCO, version 13.0, Palo Alto, CA, USA).

## 3. Results and Discussion

### 3.1. Materials’ Properties

Table 2 presents a comparative analysis of the physicochemical properties of materials, including inoculum, glucose, sugar beet pulp, wheat straw, and biochars in unmodified and chemically modified forms. The inoculum, collected from a mesophilic agricultural biogas plant, has a high moisture content of 97.8%, corresponding to a low total solids fraction. The volatile solids, which indicate the presence of degradable organic matter, account for 52.8% of the material. This study used pure glucose and sugar beet pulp to conduct the anaerobic digestion process. Glucose consists entirely of organic matter, with 100% of its total solids classified as volatile solids. In comparison, sugar beet pulp contains slightly over 10% less organic matter, which can significantly influence the anaerobic digestion process, particularly the amount of biomethane produced. Wheat straw and biochar exhibit similar moisture content and total solids, but significant differences are observed in their volatile solids. Biochar contains over 10% less organic matter than wheat straw, a reduction explained by the decomposition of organic material during pyrolysis and its volatilization. Chemical modification also impacts the degradable organic matter and ash content of biochar. Biochar treated with KOH (BC_B) shows an increase in volatile solids to 89.7% and a decrease in ash content to 10.3%. In contrast, biochar modified with H_3_PO_4_ (BC_A) exhibits a slightly lower organic matter content compared to unmodified biochar, highlighting the differing effects of chemical treatments on the material’s composition.

### 3.2. The Effect of Chemical Modification of Carbonaceous Additive Properties

The biochar made from wheat straw was chemically modified with KOH base and H_3_PO_4_ acid. The chemical modification altered the mass yield of produced carbonaceous additives, as well as their pH and surface-specific properties. The process of modification is shown in Figure 2. To facilitate process tracking, the values were recalculated and presented for 100 g of substrate. This recalculation allows for easier interpretation, as 1 g corresponds to 1% of the initial mass of the straw. Dry wheat straw underwent pyrolysis at 600 °C for 4 h, during which it was thermally decomposed into solid and gaseous products. The solid residue–biochar mass yield amounted to 21.6%, while 73.9% of pyrolytic gas was released as byproducts. The obtained biochar was then chemically treated to enhance its properties. The first batch was impregnated with 39.15 g of phosphoric acid (H_3_PO_4_), while the second batch was treated with 39.15 g of potassium hydroxide (KOH). This step allowed the biochar to absorb the chemical agents, preparing it for further processing. After chemical impregnation, the biochar was dried for 24 h to remove any excess moisture. At this stage, the weights of the chemically impregnated biochar were 31.27 g for the H_3_PO_4_-treated sample and 31.52 g for the KOH-treated sample. The chemically impregnated biochar underwent a second pyrolysis process to further enhance its surface area and porosity. This step was carried out at 500 °C for one hour, leading to further weight reductions. After this treatment, the weights of the biochar samples decreased to 28.73 g for the H_3_PO_4_ batch and 29.85 g for the KOH batch. During this step, small amounts of pyrolytic gas were also released, amounting to 2.54 g for the H_3_PO_4_ sample and 1.67 g for the KOH sample. Following the second pyrolysis, the biochar samples were thoroughly washed to remove any residual chemicals and impurities. The washing process required around 0.82 L of water per gram of biochar. This step also generated leachates and wastewater. For the H_3_PO_4_-treated biochar, 13.53 g of leachate was produced, resulting in 23.55 L of wastewater. Similarly, for the KOH-treated biochar, 10.83 g of leachate was generated, leading to 26.19 L of wastewater. The final step involved drying the washed biochar, resulting in chemically modified biochar with enhanced chemical and physical properties. The final weights of the products were 15.20 g for the H_3_PO_4_-treated biochar and 19.01 g for the KOH-treated biochar.

The overall mass yield production for biochar was 26.1%, while after chemical modification, the mass yields were 15.2% and 19.0%, respectively. The decrease in mass after chemical modification can be attributed to decomposition reactions during impregnation, activation, and washing out of organic and mineral particles during the washing stage. During activation, various chemical reactions occur. In the case of activation with KOH, the reaction begins with the melting of KOH and its reaction with the less ordered components of the charred material, generating hydrogen (H_2_) and potassium carbonate (K_2_CO_3_) [19] expanding pores [20]. Activation with H_3_PO_4_ also led to changes in the structure and porosity of the biochar, with phosphoric acid playing a role in the generation of micropores [21]. Regardless of the used reagent, the mass loss observed during chemical modification of the biochar was due to the removal of volatile components such as water, gases (CO, CO_2_, H_2_, CH_4_), and organic substances, as well as the decomposition or reaction of parts of the carbon with the activating agents. Despite the general mass loss, there was an apparent increase in the mass of the chemically modified carbon due to the retention of activating chemicals, such as potassium carbonate (K_2_CO_3_) or phosphoric acid residues. However, proper washing of the chemically modified biochar removed these residues, and the final product’s mass was lower than the mass of the raw material used (Figure 2). The yield from biochar to modified biochar was 70.4% and 88.9% for BC_A and BC_B, respectively. This is in agreement with the work of Siipola et al. [20], where the yield from biochar to activated carbon ranged from 26 to 98%. 

During the washing of the impregnated biochar, the pH changed from 1.61 to 7.11 in the case of BC_A and from 11.76 to 8.85 for BC_B. The pH of biochar was 10.8. The changes in pH during washing as a function of time are shown in Figure A1. The change in pH was caused by the washing out of impregnated activators from the biochar structure. However, when biochar after washing was measured again, the pH of BC_A and BC_B were 2.8 and 9.3. 

The use of chemicals during biochar modification led to significant changes in the functional groups present on the biochar’s surface. These changes, resulting from pyrolysis and chemical treatment, are illustrated in Figure 3a. In the FTIR analysis, the peak observed in the wheat straw sample at approximately 3300 cm^−1^ corresponds to -OH bending and water distortion. The presence of that peak is typical for hydroxyl groups, which are widely common in lignocellulosic biomass like wheat straw [22]. Also, only wheat straw presented peaks at 2800–2900 cm^−1^, which corresponds to C–H stretching, typically from aliphatic –CH_2_ or –CH_3_ groups in organic matter [23]. The most significant alteration observed in FTIR analysis during the pyrolysis of biomass is a marked decrease in the C–H and O–H reduction bands within the region of 2800–3300 cm^−1^ [24,25]. These bands were not detectable in the spectra of BC, BC_A, and BC_B. However, the aromatic C=C vibration stretching, observed in the 1600–1700 cm^−1^ range, was present in all samples. Chemical modification for both BC_A and BC_B resulted in a broader absorbance spectrum compared to non-modified biochar (BC). The thermal decomposition analysis indicates an increased presence of aromatic C=C bonds formed during pyrolysis. The spectra for BC_B, treated with KOH, are particularly intense. The alkali activation of the biochar led to a more effective degradation of volatile organic compounds and hydrogenated fragments, resulting in a higher concentration of condensed aromatic rings in the sample [26]. In contrast, H_3_PO_4_ used for BC_A promoted the formation of oxygen functional groups [27]. Peaks in the 1050–1200 cm^−1^ range were associated with the C–O–C stretching vibrations in ester groups found in cellulose and hemicellulose. Among the biochar samples, one showed the lowest peak intensity in this range, suggesting that KOH activation is more effective in breaking down oxygen-containing groups and enhancing the carbonization process. BC_A exhibited a slightly higher intensity in this region compared to BC_B, likely due to the introduction of oxygen-containing functional groups by phosphoric acid during the activation process [22,27]. Additionally, a high transmittance for aromatic C–H stretching was observed in the spectra of BC, BC_A, and BC_B at 800–900 cm^−1^ range, which was absent in the spectrum of wheat straw. The absence of this peak in the wheat straw indicates that raw biomass does not possess significant aromatic characteristics, which develop only after thermal and chemical treatments [28]. 

In summary, pyrolysis and chemical modification led to the removal of OH and C–H stretching bands at 2800–3300 cm^−1^ (aliphatic groups) while enhancing the presence of C=C stretching and introducing new C–H stretching at 800–900 cm^−1^. While it is difficult to directly link specific functional groups with their effect on the methane fermentation process, it is generally assumed that biochar, due to its redox-active surface functional groups, can enhance anaerobic digestion [29]. Redox-active groups like hydroxyl (OH), carbonyl (C=O), carbon–carbon double bonds (C=C), and carbon–oxygen single bonds (C-O) play a key role in facilitating direct interspecies electron transfer (DIET) among microbial communities, thereby enhancing methane production [30,31]. It can be observed that the intensity of regions responsible for redox-active functional groups (Figure 3b) increased after chemical modification and was the highest for BC_A. The increase in C=O compared to biochar could also be attributed to the washing process. Research by Jin et al. [31] demonstrated that washing biochar (both with water and acid) enhances electron transfer functional groups, such as ketones and quinones (carbonyl functional group), and shifts the methanogenesis pathway from hydrogenotrophic to acetoclastic [31]. 

The main effects of chemical modification can be observed in carbonaceous material’s porosity and specific surface area. The results of SSA, pore volume, and pore width (pore dimension) are presented in Table 3. The results showed that BC achieved a specific surface area (SSA_BET_) of 228 m^2^·g^−1^, a total pore volume (V_T_) of 0.115 cm^3^·g^−1^, and an average pore dimension (L) of 1.01 nm. This indicates that BC is a microporous material, which is important, especially for the adsorption of gases with low particle sizes. For BC_A, a SSA_BET_ of 219 m^2^·g^−1^ was obtained, indicating a slight reduction in this parameter due to chemical modification. The minimum increase in V_T_ of 0.128 cm^3^·g^−1^ suggests some modification of the microporous structure, and L of 1.17 nm, which is slightly larger than for BC, indicates the preservation of the microporous structure. BC_B has the highest SSA_BET_ value of 403 m^2^·g^−1^ and the highest V_T_ of 0.193 cm^3^·g^−1^, which may affect the adsorption efficiency of biomethane impurities. The value of L = 0.96 nm indicates that BC_B has micropores with smaller sizes compared to BC and BC_A. For BC_B, the S_DFT_ reached 456 m^2^·g^−1^, and the V_DFT_ increased significantly to 0.174 cm^3^·g^−1^, reflecting the formation of highly porous structures through KOH activation. The L_DFT_ value was 0.61 nm, much smaller than those of BC and BC_A, highlighting the predominance of smaller micropores. Furthermore, the V_0_ value increased to 0.156 cm^3^·g^−1^, while the L_0_ decreased to 0.70 nm, confirming the creation of extremely small micropores, which are highly advantageous for the adsorption of small molecules such as gases.

### 3.3. The Process of Anaerobic Digestion

The most biomethane was produced in the variant with glucose only (G), reaching approximately 385 mL·g_VS_^−1^ (Figure 4a). It is an interesting phenomenon that the theoretical value of biomethane produced from glucose is approximately 374 mL·g_VS_^−1^ [32]. The observed value exceeded this theoretical amount by just 2.9%, likely due to synergistic interactions with the inoculum used to initiate the process. Consequently, a greater amount of inoculum’s organic matter was converted into methane compared to the control, thereby overestimating the obtained value. The G+BC variant showed a slightly lower value, while G+BC_A and G+BC_B demonstrated the lowest biomethane production among tested conditions. The differences between the variants with biochar were minor, but G+BC_A achieved slightly higher values than G+BC_B throughout the fermentation. The difference between the maximum methane production in the control and the lowest-performing biochar variant (G+BC_B) was 6.5%. For beet pulp, the highest specific biomethane production was observed in sugar beet pulp only (BP), reaching approximately 324 mL·g_VS_^−1^ (Figure 4b). The variants with the addition of biochar (BP+BC) exhibited slightly lower yield, while BP+BC_A and BP+BC_B showed the lowest methane production among all tested conditions. The relative difference between the maximum methane production in the control and the lowest-performing biochar variant (BP+BC_B) was approximately 7.7%. The overall relative difference between maximum methane production in the glucose variant and the sugar beet pulp variant amounted to 15.6%. The lower biomethane production from BP in comparison to G is due to chemical composition. BP has more complex biomass, and not all of it can be converted by anaerobic microorganisms into methane. For both substrates, the methane production from highest to lowest was in the following order: substrate, substrate + BC, substrate + BC_A, and substrate + BC_B. The addition of carbonaceous additives indicated a potential inhibitory effect on methane production for both substrates. However, the final values differ greatly from the controls, suggesting that biochar does not completely hinder methanogenesis but may affect its efficiency negatively at analyzed process conditions and time. 

The effects of carbonaceous material addition on methane fermentation can be observed in VFA’s concentration changes (Figure 5). However, the results are not in line for both substrates. For glucose (Figure 5a), the highest concentration of acetic acid was obtained in the G+BC variant, reaching a value of approximately 1.9 g·L^−1^ on the 7th day of the process. Variants G+BC_A and G+BC_B showed slightly lower values, while the control (G) reached a maximum value at a similar level. However, on the 14th day (G) dropped lower than the variants G+BC and G+BC_A, while the G+BC_B variant dropped to around 0.2 g·L^−1^. In the case of beet pulp (BP), the highest concentration of acetic acid was recorded in the control group, reaching a value of approximately 1.3 g·L^−1^ on the 1st and 3rd days of fermentation. The variants with the addition of biochar showed slight differences, with maximum values lower than those for BP. However, on the 7th day, the experimental variants surpassed the control. In the case of propionic acid, the highest values (Figure 5b) were obtained in the control (BP) on the 3rd day, where the concentration reached approximately 0.9 g·L^−1^. Over the next few days, the values gradually decreased. Variants with glucose showed much lower values, and in most cases, the propionic acid concentration was close to zero after the 7th day of the process. 

Although the addition of carbon materials influenced the acid concentrations, the observed differences were relatively small. The type of substrate used had a more significant impact on the concentration levels. The difference between the maximum concentration of acetic acid in the variants with glucose and beet pulp was approximately 31.58%, whereas the difference in the maximum concentration of propionic acid was approximately 88.89%. Moreover, a contrasting effect of carbon materials on the concentrations of acetic and propionic acids was observed depending on the substrate type. For instance, on the third day, the addition of carbon materials led to an increase in acetic acid concentration in the glucose-based substrate, whereas a decrease was noted for the beet pulp-based substrate. However, regardless of the substrate type, the ratio of acetic acid to propionic acid was >2 in almost all cases, suggesting a lack of process imbalance. Additionally, the concentrations of acetic acid and propionic acid did not exceed 4 g·L^−1^ and 1 g·L^−1^, respectively, indicating that there was no reactor overloading or other process imbalances [33]. The process stability is further supported by the fact that the pH levels in all variants ranged from 8.4 to 9.1, remaining far from acidic conditions.

The addition of carbonaceous materials affected not only biomethane production but also biomethane production kinetics. The *B_max_* of glucose and sugar beet pulp were 396.8 and 339.6 mL·g_VS_^−1^ (Table 4). The addition of BC reduced their *B_max_* insignificantly by 1.40% and 1.67%, respectively. While the addition of BC_A and BC_B resulted in a significant decrease in *B_max_*, 5.07% and 6.28% for BC_A and BC_B, respectively. For sugar beet pulp, the addition of these materials results in a *B_max_* decrease of 4.06% and 5.30%. Generally, almost all carbon materials resulted in a small decrease in the biomethane production rate constant (*k*), except for the G+BC_A variant, where its value increased by 6.66%. Even with such an increase in the k value, the biomethane production rate (*r*) was lower than for the variant without carbon material added. The addition of carbon materials resulted in a reduction of biomethane production rates for all analyzed variants. For glucose, the biomethane production rates were reduced in the range of 0.17–7.72%, while for sugar beet pulp, the biomethane production rates were reduced in the range of 8.21–10.64%. The results are in agreement with the data shown in Figure 4. 

The results indicate that both biochar and chemically modified biochar influence the anaerobic digestion of glucose and sugar beet pulp substrates in terms of biomethane production and its kinetics. The addition of these materials slightly decreased process performance. For both substrates, biomethane production followed the same descending order: substrate alone, substrate + BC, substrate + BC_A, and substrate + BC_B. This is an interesting phenomenon because BC_B is theoretically characterized by the best properties considered for using biochar in an anaerobic digestion process. BC_B exhibited nearly twice the specific surface area and pore volume compared to both BC and BC_A (Table 3). Consequently, BC_B was expected to adsorb the highest amounts of toxic intermediates (e.g., ammonia, sulfides, excess volatile fatty acids) and provide the most habitat for anaerobic digestion microorganisms [34]. At the same time, the BC_B concentration of functional groups was intermediate (Figure 3b). Thus, the positive effects of these groups should show similar effects as BC or BC_A. 

Although many studies have shown that biochar can enhance anaerobic digestion, the obtained results in this study did not support this. For example, Ma et al. (2020) [8] observed a 25% increase in methane yield with rice husk biochar due to improved microbial colonization. Similarly, Wang et al. (2023) [11] demonstrated that magnetite-loaded biochar accelerated methanogenesis via enhanced electron transfer. Alkaline biochars have also been reported to improve pH buffering and process stability (Hu et al., 2024) [12]. However, Kozłowski et al. (2023) [13] reported that certain carbon additives can inhibit microbial activity and reduce methane yield, particularly when introduced into stable or non-stressed systems. In a performed study, the AD experiment was carried out under optimized conditions (e.g., SIR = 0.5, stable pH~8.5, low VFA < 2 mg·L^−1^), and the addition of biochar slightly decreased methane production. These findings suggest that biochar’s benefits are highly context-dependent and may only be advantageous under specific stress or imbalance conditions. The SIR of 0.50–0.25 is considered to provide enough microorganisms and buffer capacity with inoculum regarding the introduced easily degraded substrate [35]. Optimal conditions can be confirmed by the pH and acetic and propionic acid concentrations. The pH was in the alkaline range, while acid concentration (Figure 5c) did not go over problematic levels (4 and 1 g·L^−1^) [33]. As a result, methanogens were not inhibited and continued to produce biomethane effectively. However, the addition of biochar or chemically modified biochar could interfere with the anaerobic digestion of microorganisms. It is possible that these microorganisms were unable to acclimate to the presence of biochar during the testing period [36], preventing them from utilizing the additional surface area for habitat formation. Moreover, biochar may adsorb volatile fatty acids [37], making them less accessible to microorganisms and potentially decreasing methane yield. Cimon et al. 2020 [37] indicated that powdered biochar absorbed 16.6 mg of acetic acid. Other authors have suggested that biochar obtained from banana peels can effectively adsorb acetic acid, reaching up to 24.105 mmol/g [38]. The same authors also determined that the adsorption process mainly follows the Langmuir model, indicating monolayer adsorption on uniform surfaces. Other authors have also shown that acetic acid can change the chemical and physical properties of biochar [39]. Nevertheless, the adsorption of VFAs of analyzed biochars was not performed, and the measured concentration of VFAs in digestate at chosen days is not enough to confirm this theory due to the high variability in the obtained data.

## 4. Conclusions

The research aimed to study the effects of straw-derived biochar and two types of chemically modified biochar on biomethane production derived from glucose as a model substrate and sugar beet pulp as a real substrate. The use of chemical modification significantly changed biochar surface area properties and abundance in functional groups. Acid modification decreased specific surface area by 4%, while base modification increased it by 176%. The results indicate that neither the addition of biochar nor the chemically modified biochar had any positive effects on biomethane. The highest methane production was observed in reactors without biochar added, respectively, 385 and 324 mL·g_VS_^−1^ for glucose and sugar beet pulp. The difference between the maximum methane production in the control and the lowest-performing biochar variant was 6.5% for glucose and 7.7% for sugar beet pulp. The findings suggest that an anaerobic digestion process was carried out under optimal conditions (e.g., SIR = 0.5, stable pH~8.5, low VFA < 2 mg·L^−1^), and thus, biochar did not enhance methane production. In the absence of process imbalances and potentially toxic compounds such as excessive ammonia or VFAs, biochar may have absorbed compounds used by methanogens for methane production, making them less available. However, this hypothesis requires further confirmation and additional analyses of VFA adsorption on biochars and their bioavailability to methane fermentation microorganisms.

The findings of this study suggest that biochar may prove more advantageous in large-scale systems that are susceptible to process imbalances, such as ammonia inhibition or volatile fatty acid (VFA) accumulation. Future research should prioritize assessing the role of biochar in addressing these challenges and exploring its interactions with microbial communities. In practice, biochar applications should be customized to meet specific operational needs rather than being applied uniformly in biogas production.

## Figures and Tables

**Figure 1 materials-18-01608-f001:**
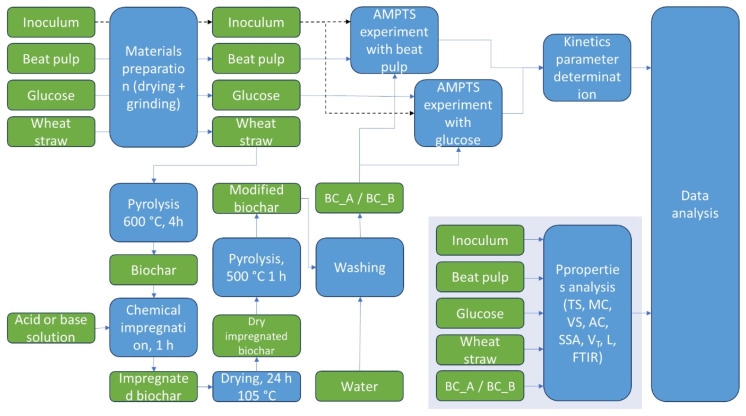
Flow diagram illustrating the experimental procedure, from materials preparation and biochar production to anaerobic digestion trials and data analysis.

**Figure 2 materials-18-01608-f002:**
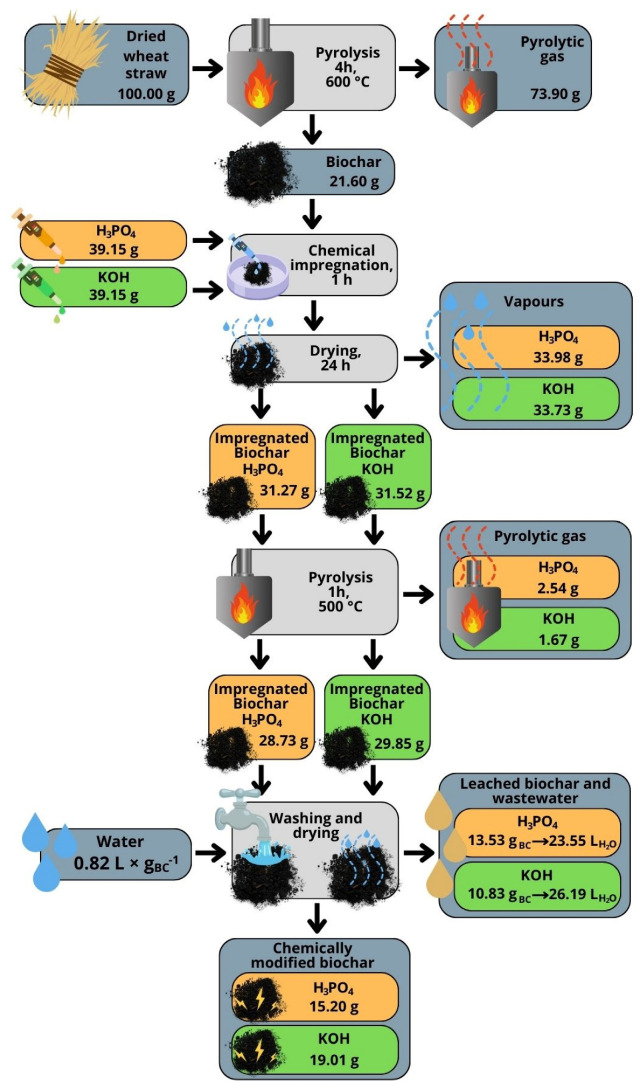
Biochar chemical modification efficiency.

**Figure 3 materials-18-01608-f003:**
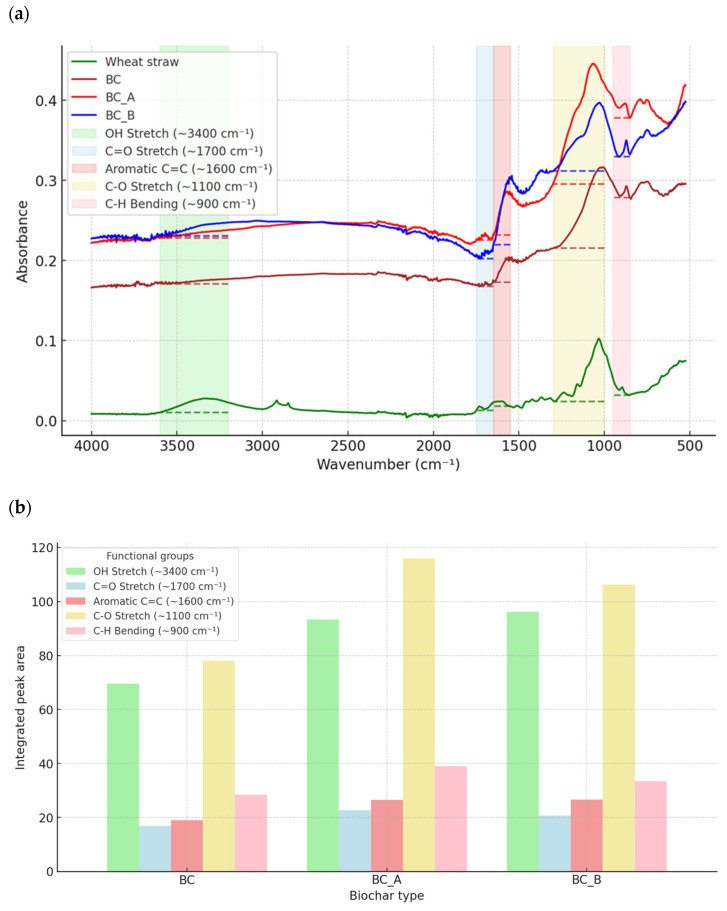
FTIR analysis. (**a**) FTIR spectra with regions considered for specific functional groups and baseline used for their integration. Dashed lines are a baseline for each functional group quantification (**b**) Integrated peak area of specific functional groups of biochars.

**Figure 4 materials-18-01608-f004:**
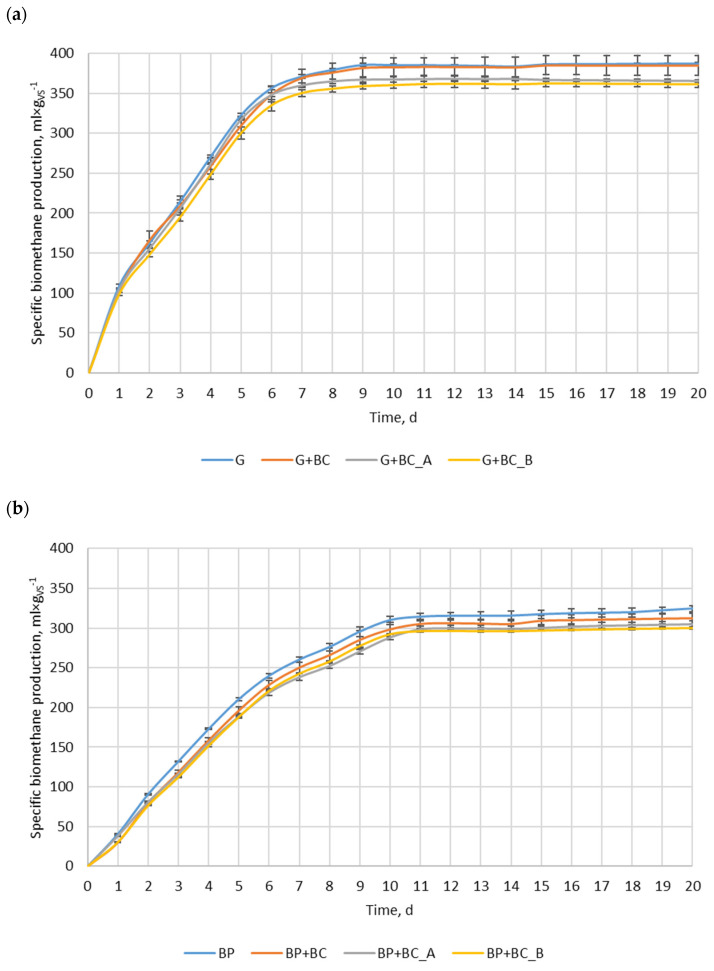
Anaerobic digestion process, (**a**) glucose, (**b**) beet pulp.

**Figure 5 materials-18-01608-f005:**
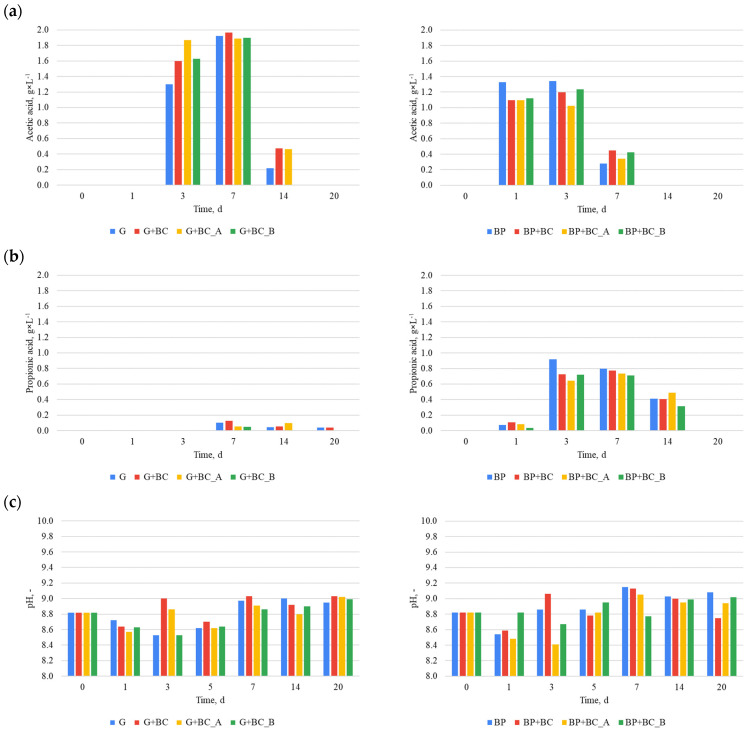
VFA concentration and pH: (**a**) acetic acid, (**b**) propionic acid, (**c**) pH.

**Table 1 materials-18-01608-t001:** Configuration of the anaerobic digestion experiment, the mass of material placed in the reactors.

Variant	I	I+G	I+G+BC	I+G+BC_A	I+G+BC_B	I+BP	I+BP+BC	I+BP+BC_A	I+BP+BC_B
Wet mass of inoculum, g	400	400	400	400	400	400	400	400	400
Dry mass of the substrate, g	0	2.30	2.30	2.30	2.30	2.81	2.81	2.81	2.81
Biochar mass, g	0	0	6.40	6.40	6.40	0	6.40	6.40	6.40

**Table 2 materials-18-01608-t002:** Basic properties of materials used in the study.

Material	MC, %	TS, %	VS, %	AC, %
Inoculum (I)	97.8 ± 0.5	2.2 ± 0.5	52.8 ± 6.2	47.2 ± 6.2
Glucose (G)	0.0 ± 0.0	100.0 ± 0.0	100.0 ± 0.0	0.0 ± 0.0
Sugar beet pulp (BP)	9.3 ± 0.1	90.7 ± 0.1	89.8 ± 2.2	10.2 ± 2.2
Wheat straw (WS)	5.8 ± 0.1	94.2 ± 0.1	94.6 ± 0.2	5.4 ± 0.2
Biochar (BC)	4.7 ± 0.3	95.3 ± 0.3	83.3 ± 0.2	16.7 ± 0.2
Biochar modified with KOH base (BC_B)	7.4 ± 0.1	92.6 ± 0.1	89.7 ± 0.1	10.3 ± 0.1
Biochar modified with H_3_PO_4_ acid (BC_A)	6.7 ± 0.4	93.3 ± 0.4	81.7 ± 0.7	18.3 ± 0.7

**Table 3 materials-18-01608-t003:** Surface properties of carbonaceous additives.

Material, -	SSA_BET_, m^2^·g^−1^	V_T_, cm^3^·g^−1^	L, nm	SSA_DFT_, m^2^·g^−1^	V_DFT_, cm^3^·g^−1^	L_DFT_, nm	V_0_, cm^3^·g^−1^	L_0,_ nm
BC	228	0.115	1.01	213	0.108	1.68	0.071	1.19
BC_A	219	0.128	1.17	238	0.118	1.68	0.077	1.05
BC_B	403	0.193	0.96	456	0.174	0.61	0.156	0.70

**Table 4 materials-18-01608-t004:** Kinetics parameter of biomethane production.

Variant	*B_max_*, mL·g_VS_^−1^	*k*, d^−1^	*r*, mL·(g_VS_·d)^−1^	*R*^2^, -
G	396.8 ± 0.6	0.30 ± 0.00	119.1 ± 1.7	0.96
G+BC	395.4 ± 14.4	0.29 ± 0.01	115.4 ± 6.5	0.96
G+BC_A	376.7 ± 2.9	0.32 ± 0.00	118.9 ± 0.9	0.96
G+BC_B	371.9 ± 4.3	0.30 ± 0.01	109.9 ± 4.1	0.96
BP	339.6 ± 7.3	0.19 ± 0.00	65.8 ± 0.6	0.98
BP+BC	333.9 ± 8.6	0.18 ± 0.00	60.4 ± 2.3	0.97
BP+BC_A	325.8 ± 5.0	0.18 ± 0.00	58.2 ± 1.0	0.98
BP+BC_B	321.6 ± 1.9	0.18 ± 0.00	58.8 ± 0.5	0.97

Tukey test probability values at *p*< 0.05 are presented in Table A1, Table A2, Table A3 and Table A4. The variants with glucose and sugar beet pulp were analyzed separately.

## Data Availability

All experimental data were compiled into a dataset and submitted to the Knowledge Base repository of UPWr, where they are available under the DOI number: http://dx.doi.org/10.57755/ntsk-bh25 (accessed on 5 March 2025).

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
