# Peer review of "The Effects of Chemically Modified Biochar on Biomethane Production from Glucose and Sugar Beet Pulp"

_materials, 2025, doi:10.3390/ma18071608_

Round 1
Reviewer 1 Report
Comments and Suggestions for Authors
The paper studied the effects of straw-derived biochar and two types of chemically modified biochar on biomethane production from glucose as a model substrate and sugar beet pulp as a real substrate. The authors conducted a series of experimental studies and characterizations, and conducted a detailed analysis. The following suggestions need to be addressed to improve the paper.
- Please explain the experimental method by introducing the content shown in Figure 1.
- In the "Introduction" section, please explain the novelty of the research work.
- The “table 1”mentioned in the first paragraph of section 3.1 should be modified to “table 2”.
- In section 3.2, please provide an explanation for Figure 2.
- Please add some quantitative analysis results to the conclusion.
Reviewer 2 Report
Comments and Suggestions for Authors
This manuscript presents an investigation into the effects of straw-derived biochar and two chemically modified biochars on biomethane production from glucose and sugar beet pulp.
- The authors should further discuss how their findings compare with existing literature, particularly in cases where biochar has been found to enhance AD efficiency.
- The conclusion should emphasize how these findings impact future research and potential practical applications.
- The novelty of the work should be highlighted better.
- The authors should consider including Thermogravimetric Analysis (TGA) measurements to better characterize the thermal stability and decomposition behavior of the biochar before and after modification. This would provide additional insight into its physicochemical properties and its potential role in anaerobic digestion.
- The results suggest that biochar did not improve biomethane production and might have adsorbed volatile fatty acids (VFAs). Can the authors provide additional justification for this hypothesis?
moderate revision
Reviewer 3 Report
Comments and Suggestions for Authors
Dear authors,
This manuscript provides significant insight into conducting a systematic evaluation of the effects of straw-derived biochar and two types of modified biochar on biomethane production derived from glucose and sugar beet pulp. The subject covered in this article will interest Journal Materials readers but may be accepted for publication only after minor revision. Here are my comments:
Comment 1: English correction is required.
Comment 2: Lines 156-158 – …,, The following wavenumber ranges were considered: OH 3200–3600 cm⁻¹; C=O 1650–1750 cm⁻¹; C=C 1550–1650 cm⁻¹; C-O 1000–1300 cm⁻¹, and C-H Bending 850–950 cm⁻¹…” – Please provide references that support each of the listed values for wavenumber ranges.
Comment 3: Line 199 – …,, Samples were collected on days 1, 3, 7, 14, 20…” – Please specify the number of samples in question. Based on your statement, does it mean that you only had 5 samples or that you had multiple batches of samples and used the best or average values from them? Please provide us with that explanation in the text.
Comment 4: Lines 317-345 – Please complete the text with an explanation about peaks observed in the range of ⁓2300-1900 cm-1.
Comment 5: Line 411 – Figure 1– Please mark the images with (a) and (b), so you are missing the brackets on the left side and if you are able, please separate these curves a little to make them easier to see or put some brighter colors.
Comment 6: Line 444 – Figure 4 – Within Figure 4, you have 6 images that you need to label each separately with (a), (b)....(f) and provide an explanation for each of them in the text below, and put the word Figure 1 in bold.
Comment 7: Line 490 – Please add in the conclusion why large-scale biogas production is important and where it would be useful with a clear explanation of the scientific motivation.
Comment 8: The references listed in the manuscript are mostly recent, relevant publications (within the last 5 years). Self-citations are present and consistent with the text.

Reviewer 4 Report
Comments and Suggestions for Authors
Dear authors
The study investigates the impact of the addition of chemically modified biochar on the production of biomethane through two substrates: glucose and beet pulp. The study addresses a relevant topic for optimizing anaerobic digestion and the valorization of biochar as an additive.
In the introduction, the justification for selecting chemical modifiers (H₃PO₄ and KOH) was not well founded.
Some statements do not have direct references; the authors should again note lines that were not correctly referenced.
The study mentions that biochar can adsorb VFAs, but no direct measurements validate this hypothesis. Since the authors raise the hypothesis but do not perform experiments to prove it, an analysis of VFA adsorption by biochar could strengthen the interpretation of the results.
The biogas quantification method could include more details on equipment calibration and accuracy, such as the AMPTS® II typically requires calibration with a standard gas (e.g., pure methane or a known CH₄/CO₂ mixture).
Improve the discussion of the results, including more explanations of the observed effects, such as analysis of the possible effects of VFA adsorption, the impact of biochar on methanogenic microbiota, and whether the experimental conditions favored methanogenesis without the need for additives.
The paper suggests that biochar may have adsorbed VFAs, reducing their availability for methanogenesis, but does not present direct measurements to support this claim. Therefore, if the authors cannot perform these analyses, they should clarify in the discussion that this hypothesis was not tested experimentally.
Round 2
Reviewer 2 Report
Comments and Suggestions for Authors
I recommend the publication of this article
Comments on the Quality of English Languagemoderate revision
Reviewer 4 Report
Comments and Suggestions for Authors
The corrections made by the authors made the article suitable for publication. I consider the manuscript accepted.